# Learning Implicit Generative Models by Teaching Explicit Ones

## Abstract

Implicit generative models are difficult to train as no explicit probability density functions are defined. Generative adversarial nets (GANs) propose a minimax framework to train such models, which suffer from mode collapse in practice due to the nature of the JS-divergence. In contrast, we propose a *learning by teaching* (LBT) framework to learn implicit models, which intrinsically avoid the mode collapse problem because of using the KL-divergence. In LBT, an auxiliary explicit model is introduced to *learn* the distribution defined by the implicit model while the later one's goal is to *teach* the explicit model to match the data distribution. LBT is formulated as a bilevel optimization problem, whose optimum implies that we obtain the maximum likelihood estimation of the implicit model. We adopt an unrolling approach to solve the challenging learning problem. Experimental results demonstrate the effectiveness of our method.

## 1 Introduction

Deep generative models (Kingma & Welling, 2013; Goodfellow et al., 2014; Oord et al., 2016) have the ability to capture the distributions over complicated manifolds, e.g., natural images. Most recent state-of-the-art deep generative models (Radford et al., 2015; Arjovsky et al., 2017; Karras et al., 2017) are usually implicit statistical models (Mohamed & Lakshminarayanan, 2016), also called implicit probability distributions. Implicit distributions are flexible as they are specified by a sampling procedure rather than a tractable density. However, this implicit nature makes them difficult to train since maximum likelihood estimation (MLE) is not directly applicable.

Generative adversarial networks (GANs) (Goodfellow et al., 2014) address this difficulty by adopting a minimax game, where a discriminator $D$ is introduced to distinguish whether a sample is real (from the data distribution) or fake (from a generator $G$), while $G$ tries to fool $D$ via generating realistic samples. In practice, $G$ and $D$ are optimized alternatively and GANs suffer from the mode collapse problem. This is because $G$ is optimized to generate samples which are considered as real ones with high confidence by the current $D$, and won't be penalized for missing modes in data distribution. Although various methods (Metz et al., 2016; Zhao et al., 2016; Arjovsky et al., 2017) try to modify the vanilla GANs to alleviate the problem, they are still formulated in the minimax framework and do not address the intrinsic weakness of GANs. Therefore, the mode collapse problem of training implicit models is still largely open.

This problem mainly arises from the objective function in GAN, i.e., JS-divergence over data distribution $p$ and the generator distribution $p_G$, which is more tolerant to mode collapse compared to the KL-divergence. As illustrated in the example in Fig. 1, local optima with mode collapse can be found by optimizing JS-divergence whereas $KL(p||p_G)$ achieves its optima iff $p = p_G$. To address this issue, we propose a novel teacher-student framework to optimize the KL-divergence where we *learn* an implicit generator $G$ (also referred as a teacher) by *teaching* a likelihood estimator $E$ (also referred as a student) to match the data distribution. In particular, the training scheme is as follows:

 (a) The student $E$ is trained to maximize the log-likelihood of samples generated by the teacher $G$.

 (b) The student $E$ is evaluated on the real data samples in terms of log-likelihood as well, and the teacher $G$ is trained to improve such log-likelihood based on the signal from $E$.

According to the scheme, we refer to our framework as *learning by teaching* (LBT).

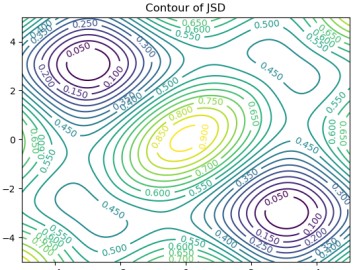 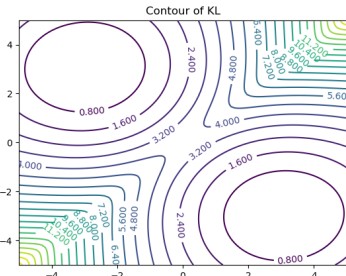

Figure 1: Suppose the data distribution is a mixture of Gaussian (MoG), i.e., $p(x) = 0.5\mathcal{N}(-3, 1) + 0.5\mathcal{N}(3, 1)$, and the model distribution is a MoG with learnable means $a$ and $b$, i.e., $p_G(x) = 0.5\mathcal{N}(a, 1) + 0.5\mathcal{N}(b, 1)$. The figure demonstrate the contour of the two divergences with the x-axis and y-axis representing the value of $a$ and $b$ respectively. The JS-divergence allows a collapsed local optima with $a = -3.25, b = -2.69$ (numerical results).

Intuitively, in LBT, $E$ tells $G$ what must be generated whereas in GAN, $D$ tells $G$ what cannot be generated. Specifically, in LBT, $E$ learns $p_G$ and $G$ aims to adjust its distribution to maximize the log likelihood of real samples evaluated by the learned $E$. If $p_G$ misses some modes, then $p_E$ on these modes is low, and $G$ will be penalized heavily. In other words, in LBT, the estimator directs the generated samples to overspread the support of data distribution. In GAN, the goal of $D$ is to distinguish whether a sample is real or fake and the goal of $G$ is to fool $D$ by generating realistic samples. $G$ will be penalized much more heavily when generating fake patterns than when missing modes. Therefore, we can say that LBT and GAN are complementary to each other, where $E$ helps $G$ to overspread the data distribution and $D$ helps $G$ to generate realistic samples. Thus, we propose the combined LBT-GAN to improve the performance.

Formally, LBT is formulated as a bilevel optimization (Colson et al., 2007) problem, where an *upper* level optimization problem (step (b)) is dependent on the optimal solution of a *lower* level problem (step (a)). The gradients of the upper problem w.r.t. the parameters of $G$ are unknown since the optimal solution of $E$ cannot be analytically expressed by $G$'s parameters. Though the influence function (Koh & Liang, 2017) provides a principle way to differentiate through the bilevel optimization problem, it is computationally expensive. Instead, we propose to use the unrolling technique (Metz et al., 2016) to efficiently obtain approximate gradients, which are closely connected to the exact ones given by the influence function. Theoretically, under non-parametric conditions, LBT ensures that both of the teacher $G$ and the student $E$ converge to the data distribution (See proof in Sec. 4). Besides, we provide further empirical analysis of the case where a $E$ with limited capbility can still help $G$ to resist to mode collapse in Appendix B.

To summarize, our contributions are threefold:

1. We propose a novel framework LBT to train an implicit generator by teaching a density estimator. LBT intrinsically avoids the mode collapse problem and is fully compatible with GANs.

2. Theoretically, we prove that the implicit model will converge to the data distribution in both LBT and LBT-GAN.

3. Empirically, we show the effectiveness of LBT and LBT-GAN on both synthetic and real datasets.

## 2 RELATED WORK

Implicit statistical models (Mohamed & Lakshminarayanan, 2016) are of great interests with the emergence of GANs (Goodfellow et al., 2014) which introduces a minimax framework to train implicit generative models. Nowozin et al. (2016) generalize the original GANs via introducing a broad class of $f$-divergence for optimization. In comparison, LBT provides a different way to optimize the KL-divergence and achieves good results on avoiding the mode collapse problem and generating samples when combined with GAN. Arjovsky et al. (2017) propose to minimize the earth mover's distance to avoid the problem of gradient vanishing in vanilla GANs. Besides, Li et al. (2015) train implicit models by matching momentum between generated samples and real samples.

Mode collapse is a well-known problem in practical training of GANs. Much work has been done to analyze and alleviate the problem (Arjovsky & Bottou, 2017; Arjovsky et al., 2017; Mao et al., 2017; Metz et al., 2016; Srivastava et al., 2017). Unrolled GAN (Metz et al., 2016) propose to unroll the update of the discriminator in GANs. The unrolling helps capturing how the discriminator would react to a change in the generator. Therefore it reduces the tendency of the generator to collapse all

samples into a single mode. Srivastava et al. (2017) propose VEEGAN that introduces an additional reconstructor net to map data back to the noise space. Then a discriminator on the joint space is introduced to learn the joint distribution, similar as in ALI (Dumoulin et al., 2016). Lin et al. (2017) propose to modify the discriminator to distinguish whether multiple samples are real or generated. Though such methods can resist mode collapsing to some extent, they are still restricted to the minimax formulation, which makes the training extremely unstable. We directly compare LBT with existing methods (Metz et al., 2016; Srivastava et al., 2017; Lin et al., 2017) in our experiments.

In LBT, we need to evaluate the influence of the training data of the estimator, i.e., generated samples, on the likelihood of test data, i.e., real samples, which is closely related to the influence function methods. Koh & Liang (2017) propose to use influence function to model the affect of training data on the loss of test data, which is equivalent to a quadratic approximation at the optimal point. Zhang et al. (2018) apply this method to debug the training data using a set of trusted items, where the authors presume that there is bias in the training data. Our method can be treated as another instance of the influence function , where we try to learn an implicit generative model using the influence of the generated samples on the log-likelihood of the real data evaluated by the estimator.

## 3  METHOD

Consider an implicit model (or a generator) $G(\cdot; \theta)$ parameterized by $\theta$ that maps a simple random variable $z \in \mathbb{R}^H$ to a sample $x$ in the data space $\mathbb{R}^D$, i.e., $x = G(z; \theta)$. Here, $z$ is typically drawn from a standard Gaussian distribution $p_Z$ and $G$ is typically a feed-forward neural network. The sampling procedure defines a distribution over the data space, denoted as $p_G(x; \theta)$. Our goal is to train the generator to approximate the data distribution $p(x)$.

Since the generator distribution is implicit, it is infeasible to adopt maximum likelihood estimation directly to train the generator. To address the problem, we propose learning by teaching (LBT), which introduces an auxiliary density estimator $p_E(x; \phi)$ parameterized by $\phi$ ( e.g., a VAE (Kingma & Welling, 2013)) to fit the generator distribution $p_G(x; \theta)$ by maximizing the log-likelihood on generated samples. We train the generator to maximize the estimator's likelihood evaluated on the training data. Formally, LBT is defined as a bilevel optimization problem (Colson et al., 2007):

$$
\begin{aligned}
\max_{\theta} \quad & \mathbb{E}_{x \sim p(x)}[\log p_E(x; \phi^\star(\theta))], \\
\text{s.t.} \quad & \phi^\star(\theta) = \arg\max_{\phi} \mathbb{E}_{z \sim p_Z}[\log p_E(G(z; \theta); \phi)],
\end{aligned}
\tag{1}
$$

where $\phi^\star(\theta)$ clarifies that the optimal $\phi^\star$ depends on $\theta$. For notational convenience, we denote

$$
\begin{aligned}
f_G(\phi^\star(\theta)) &= \mathbb{E}_{x \sim p(x)}[\log p_E(x; \phi^\star(\theta))], \\
f_E(\theta, \phi) &= \mathbb{E}_{z \sim p_Z}[\log p_E(G(z; \theta); \phi)],
\end{aligned}
\tag{2}
$$

in the sequel. In Sec. 4, we provide theoretical analysis to show that the generator distribution can converge to the data distribution given that the generator and the estimator have enough capacity.

### 3.1  DIFFERENTIATE THROUGH THE BILEVEL OPTIMIZATION PROBLEM

The bilevel problem is generally challenging to solve. Here, we present a stochastic gradient ascend algorithm (i.e., Algorithm 1) by using an unrolling technique to derive the gradient. Specifically, to perform gradient ascend, we calculate the gradient of $f_G$ with respect to $\theta$ as follows:

$$
\frac{\partial f_G(\phi^\star(\theta))}{\partial \theta} = \frac{\partial f_G(\phi^\star(\theta))}{\partial \phi^\star(\theta)} \frac{\partial \phi^\star(\theta)}{\partial \theta} = \frac{\partial f_G(\phi^\star(\theta))}{\partial \phi^\star(\theta)} \int_z \frac{\partial \phi^\star(\theta)}{\partial G(z; \theta)} \frac{\partial G(z; \theta)}{\partial \theta} p_Z dz,
\tag{3}
$$

where both $\frac{\partial f_G(\phi^\star(\theta))}{\partial \phi^\star(\theta)}$ and $\frac{\partial G(z; \theta)}{\partial \theta}$ are easy to calculate. However, $\frac{\partial \phi^\star(\theta)}{\partial G(z; \theta)}$ is intractable since $\phi^\star(\theta)$ can not be expressed as an analytic function of the generated samples $G(z; \theta)$. In the following, we rewrite $G(z; \theta)$ as $x_z$ and $\frac{\partial f_E(\theta, \phi)}{\partial \phi}$ as $\nabla \phi$ for simplicity.

On one hand, the influence function (Koh & Liang, 2017) provides a way to calculate the gradient of $\phi^\star(\theta)$ w.r.t. the generated samples $x_z$ as follows:

$$
\frac{\partial \phi^\star(\theta)}{\partial x_z} = -H_{\phi^\star}^{-1} \nabla_{x_z} \left( \left. \frac{\partial f_E(\theta, \phi)}{\partial \phi} \right|_{\phi^\star} \right),
\tag{4}
$$

---

**Algorithm 1** Stochastic Gradient Ascend Training of LBT with the Unrolling Technique

---

**Input:** data $x$, learning rate $\eta_\theta$ and $\eta_\phi$, unrolling steps $K$ and inner update iterations $M$.
Initialize parameters $\theta_0$ and $\phi_0$, and $t = 1$.
**repeat**
    $\phi_t^0 \leftarrow \phi_{t-1}$
    **for** $i = 1$ **to** $M$ **do**
        $\phi_t^i \leftarrow \phi_t^{i-1} + \eta_\phi \cdot \left. \frac{\partial f_E(\theta,\phi)}{\partial \phi} \right|_{\phi_t^{i-1}}$
    **end for**
    Update $\phi$: $\phi_t \leftarrow \phi_t^M$
    $\phi_t^0 \leftarrow \phi_t$
    Unrolling: $\phi_t^K \leftarrow \phi_t^0 + \sum_{i=1}^K \eta_\phi \cdot \left. \frac{\partial f_E(\theta,\phi)}{\partial \phi} \right|_{\phi_t^{i-1}}$
    Update $\theta$: $\theta_t \leftarrow \theta_{t-1} + \eta_\theta \frac{\partial f_G(\phi_t^K)}{\partial \theta}$
    Update $t$: $t \leftarrow t + 1$
**until** Both $\theta$ and $\phi$ converge.

---

where $H_{\phi^\star}$ is the Hessian of the objective $f_E$ w.r.t. $\phi$ at $\phi^\star$ and is negative semi-definite (Koh & Liang, 2017). However, calculating the Hessian and its inverse is computationally expensive.

On the other hand, a local optimum $\hat{\phi}^\star$ of the density estimator parameters can be expressed as the fixed point of an iterative optimization procedure,

$$
\begin{aligned}
\phi^0 &= \phi \\
\phi^{k+1} &= \phi^k + \eta \cdot \left. \frac{\partial f_E(\theta,\phi)}{\partial \phi} \right|_{\phi^k} \\
\hat{\phi}^\star &= \lim_{k \to \infty} \phi^k,
\end{aligned}
\tag{5}
$$

where $\eta$ is the learning rate[1]. Since the samples used to evaluate the likelihood $f_E(\theta,\phi)$ are generated by $G$, each step of the optimization procedure is dependent on $\theta$. We thus write $\phi^k(\theta,\phi^0)$ to clarify that $\phi^k$ is a function of $\theta$ and the initial value $\phi^0$. Since $\frac{\partial f_E(\theta,\phi)}{\partial \phi}$ is differentiable w.r.t. $x_z$ for most density estimators such as VAEs, $\phi^k(\theta,\phi^0)$ is also differentiable w.r.t. $x_z$. By unrolling for $K$ steps, namely, using $\phi^K(\theta,\phi^0)$ to approximate $\phi^\star(\theta)$ in the objective $f_G(\phi^\star(\theta))$, we optimize a surrogate objective for the generator formulated as $f_G(\phi^K(\theta,\phi^0))$. Thus, the term $\frac{\partial \phi^\star(\theta)}{\partial x_z}$ is approximated as $\frac{\partial \phi^\star(\theta)}{\partial x_z} \approx \frac{\partial \phi^K(\theta,\phi^0)}{\partial x_z}$, which is known as the unrolling technique (Metz et al., 2016).

We now build connections between the above approximate gradients and the exact gradients given by the influence function. Assuming that the parameters of the density estimator is at its optimum $\phi^\star$, i.e., $\phi^0 = \phi^\star$, we examine the case of $K = 1$. The result of one step unrolling is given by:

$$
\frac{\partial \phi^1}{\partial x_z} = \frac{\partial(\phi^0 + \eta \nabla \phi)}{\partial x_z} = \eta \frac{\partial \nabla \phi}{\partial x_z} = \eta \nabla_{x_z} \left( \left. \frac{\partial f_E(\theta,\phi)}{\partial \phi} \right|_{\phi^0} \right) = \eta \nabla_{x_z} \left( \left. \frac{\partial f_E(\theta,\phi)}{\partial \phi} \right|_{\phi^\star} \right).
\tag{6}
$$

Note that the inner product of $\frac{\partial \phi^1}{\partial x_z}$ and $\frac{\partial \phi^\star}{\partial x_z}$ given by the influence function is positive because the Hessian $H_{\phi^\star}$ is negative semi-definite (Koh & Liang, 2017). Therefore, the unrolling technique essentially gives an approximation of the influence function in Eqn. (4) under the condition that the estimator is good enough (near to its optimaum $\phi^\star$). Besides, the unrolling technique is much more efficient as it does not need to inverse the Hessian matrix.

Finally, the generator and the likelihood estimator can be updated using the following process,

$$
\theta \leftarrow \theta + \eta_\theta \frac{\partial f_G(\phi^K(\theta,\phi))}{\partial \theta}, \quad \phi \leftarrow \phi + \eta_\phi \frac{\partial f_E(\theta,\phi)}{\partial \phi},
\tag{7}
$$

where $\eta_\theta$ and $\eta_\phi$ are the learning rates for the generator and the estimator, respectively. We perform several updates of $\phi$ per update of $\theta$ to keep the estimator good. Note that for other gradient-based optimization methods such as Adam (Kingma & Ba, 2014), the unrolling procedure is similar (Metz et al., 2016). In our experiments, only a few steps of unrolling, e.g., 5 steps, are sufficient for the training. The whole training algorithm is described in Algorithm 1.

---

[1]We have omitted the learning rate decay for simplicity.

### 3.2 Augmenting LBT with a Discriminator

In LBT, the uncovered modes give a large penalty to the generator $G$ through the unrolled estimator $E$ and $E$ can successfully spread the generated samples to match the whole data distribution. However, due to the zero-avoiding property of MLE (Nasrabadi, 2007), it can hardly give a large penalty to the generator for generating low-quality samples when all modes are covered by the generated samples. Hence, we propose to augment the LBT framework by incorporating a discriminator to penalize the generator for generating unreal samples. Formally, the objective is formulated as follows:

$$
\begin{aligned}
\max_{\theta} \quad & \mathbb{E}_{x \sim p(x)}[\log p_E(x; \phi^\star(\theta))] + \lambda_G \mathbb{E}_{z \sim p_Z}[\log D(G(z; \theta); \psi^\star)], \\
\text{s.t.} \quad & \phi^\star(\theta) = \arg\max_{\phi} \mathbb{E}_{z \sim p_Z}[\log p_E(G(z; \theta); \phi)], \\
& \psi^\star = \arg\max_{\psi} \mathbb{E}_{x \sim p(x)}[\log D(x; \psi)] + \mathbb{E}_{z \sim p_Z}[\log(1 - D(G(z; \theta); \psi))],
\end{aligned}
\tag{8}
$$

where $\psi$ is the parameters of discriminator $D$ and $\lambda_G$ balances the weight between two losses. We call the above method LBT-GAN.

## 4 Theoretical Analysis

In this section, we prove that both the generator and the estimator can converge to the data distribution, under the assumption that the generator and estimator have infinity capacity.

**Theorem 1.** *For a fixed generator $G$, the optimal likelihood estimator $E$ converges to the generator distribution, i.e., $p_E(x; \phi^\star) = p_G(x; \theta)$.*

*Proof.* The objective of the estimator is to maximize the log-likelihood of the generated samples:

$$
\begin{aligned}
\mathbb{E}_{x \sim p_G(x;\theta)}[\log p_E(x; \phi)] = & \mathbb{E}_{x \sim p_G(x;\theta)}[\log \frac{p_E(x; \phi)}{p_G(x; \theta)}] + \mathbb{E}_{x \sim p_G(x;\theta)}[\log p_G(x; \theta)] \\
= & - KL(p_G(x;\theta)||p_E(x;\phi)) - H(p_G(x;\theta)),
\end{aligned}
\tag{9}
$$

where $H(p_G(x;\theta))$ is the entropy of the generator distribution which is a constant with respect to the estimator $E$. Hence, maximizing Eqn. (9) is equivalent to minimizing the KL divergence between $p_G$ and $p_E$. The likelihood estimator thus achieves optimal when $p_E = p_G$. $\square$

**Theorem 2.** *Maximizing Eqn. (1) is equivalent to minimizing the KL-divergence between the data distribution and the generator distribution.*

*Proof.* Because $p_E(x; \phi^\star) = p_G(x; \theta)$ (proved above), it is straightforward that maximizing Eqn. (1) is equivalent to solving the problem: $\max_{\theta} \mathbb{E}_{x \sim p(x)}[\log p_G(x; \theta)]$, which is the maximum likelihood estimation and is equivalent to minimizing the KL-divergence between $p(x)$ and $p_G(x; \theta)$. The optimal is achieved when $p_G = p$. $\square$

The conclusions of the above two theorems imply that the global optimum of LBT is achieved at $p_G = p_E = p$. Since the optimum of GAN's minimax framework is also achieved at $p_G = p$, it is straight-forward that LBT-GAN in Eqn. (8) has the same optimal solution as LBT.

## 5 Experiments

We now present the empirical results of LBT on both synthetic and real datasets. Throughout the experiments, we set the unrolling steps $K = 5$ and use Adam (Kingma & Ba, 2014) optimization method with the default setting for both the generator and the estimator (and the discriminator for LBT-GAN). We set the inner update iterations $M$ for the estimator to 15. We use variational auto-encoders (VAEs) as the density estimators for both LBT and LBT-GAN. All the decoders and encoders in VAEs are two-hidden-layer fully-connected MLPs. In LBT-GAN, we set $\lambda_G = 0.1$ for synthetic datasets and $\lambda_G = 1$ for real datasets. We will release the code after the blind review.

### 5.1 Synthetic Datasets

We first compare LBT with state-of-the-art competitors (Goodfellow et al., 2014; Mao et al., 2017; Metz et al., 2016; Srivastava et al., 2017) on 2-dimensional (2d) synthetic datasets, which are convenient for qualitative and quantitative analysis.

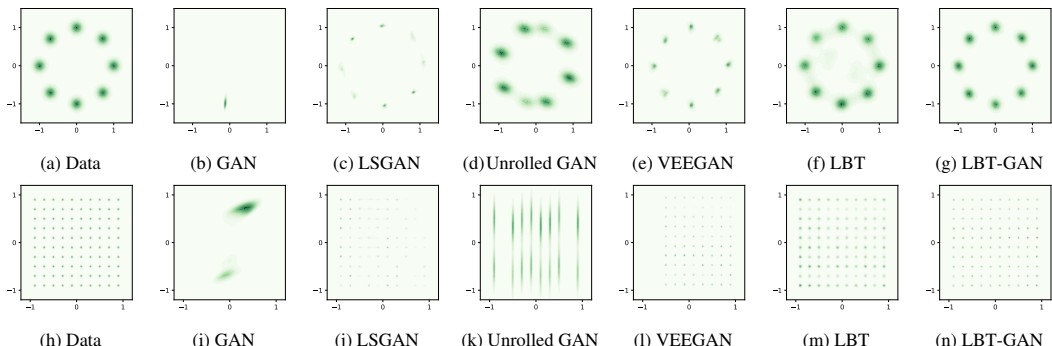

Figure 2: Density plots of the true distributions and the generator distributions of different methods trained on the ring data (Top) and the grid data (Bottom).

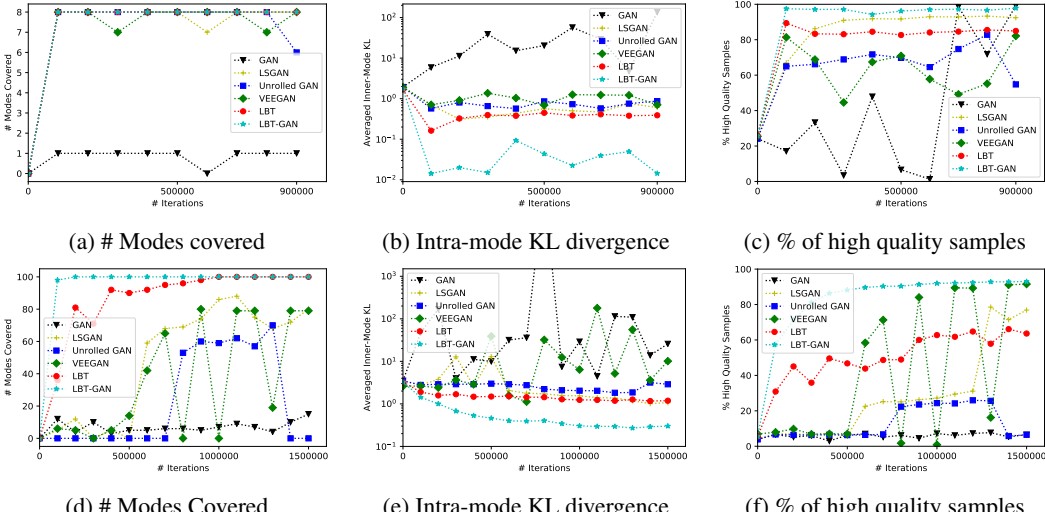

Figure 3: Three different metrics evaluated on the generator distributions of different methods trained on the ring data (Top) and the grid data (Bottom). The metrics from left to right are: Number of modes covered (the higher the better); Averaged intra-mode KL divergence (the lower the better); Percentage of high quality samples (the higher the better).

Specifically, we construct two datasets: (i) **ring**: mixture of 8 2d Gaussian distributions arranged in a ring and (ii) **grid**: mixture of 100 2d Gaussian distributions arranged in a 10-by-10 grid. All of the mixture components are isotropic Gaussian, i.e., with standard diagonal covariance. For the ring data, the deviation of each Gaussian component is $\text{diag}(0.1, 0.1)$ and the radius of the ring is 1 [2]. For the grid data, the spacing between adjacent modes is $0.2$ and the deviation of each Gaussian component is $\text{diag}(0.01, 0.01)$. Fig. 2a and Fig. 2h show the true distributions of the ring data and the grid data, respectively. For fair comparison, we use the same network architectures (two-hidden-layer fully-connected MLPs) for the generators of all methods. For GAN-based methods, the discriminators are also two-hidden-layer fully-connected MLPs. The number of hidden units for the generators and the estimators (and the discriminators for LBT-GAN) is 128.

To quantify the quality of the generator learned by different methods, we report 3 metrics to demonstrate different characteristics of generator distributions:

- **Percentage of High Quality Samples (Srivastava et al., 2017)**: We define a generated sample as a *high quality* sample if it is within three standard deviations of the nearest mode. We generate $500,000$ samples from each method and report the percentage of high quality samples.

- **Number of Modes Covered**: We count a mode as a *covered mode* if the number of its high quality samples is greater than $20\%$ of the expected number. For example, when we generate $500,000$

---

[2]In the original Unrolled GANs setting (Metz et al., 2016), the std of each component is 0.02 and the radius of the ring is 2. In our setting, the ratio of std to radius is 10 times larger. We choose this setting in order to characterize different performance of "Intra-mode KL divergence" clearly.

Table 1: Degree of mode collapse measured by number of mode captured and KL divergence on Stacked MNIST. Results are averaged over 5 runs.

|  | DCGAN | ALI | Unrolled GAN | VEEGAN | DCGAN(ours) | LBT-GAN |
|---|---|---|---|---|---|---|
| Modes | 99 | 16 | 48.7 | 150 | 188.8 | **999.6** |
| KL | 3.4 | 5.4 | 4.32 | 2.95 | 3.17 | **0.19** |

(a) D=1 size of G: DCGAN(left) and LBT-GAN(right)    (b) D=0.5 size of G:DCGAN(left) and LBT-GAN(right)

Figure 4: Generated samples of DCGANs and LBT-GANs with different size of discriminators. LBT-GANs can successfully generate high quality samples under different network architectures.

samples from the generator trained on the ring dataset (which has 8 modes), the expected number of high quality samples for each mode is about $500,000/8 = 62,500^3$. Thus in this case, we count a mode as covered if it has at least $62,500 \times 20\% = 12,500$ high quality samples. Intuitively, lower number of modes covered indicates higher global mismatch between the generator distribution and the true distribution.

- **Averaged Intra-Mode KL Divergence**: We assign each generated sample to the nearest mode of the true distribution. Then for each mode, we fit a Gaussian model on all assigned samples. The fitted Gaussian model can be viewed as an estimation of the generator distribution at the corresponding mode, whose true distribution is also Gaussian. We analytically calculate the KL divergence between the true distribution and the estimated distribution at each mode, which we call *intra-mode KL divergence*. Intuitively, the intra-mode KL divergence measures the local mismatch between the generator distribution and the true distribution. We report the averaged intra-mode KL divergence over all modes.

Fig. 2 shows the generator distributions learned by different methods. Each distribution is plotted using kernel density estimation with $500,000$ samples. We can see that our LBT manages to cover the largest number of modes on both ring and grid datasets compared to other methods, demonstrating that LBT can generate globally diverse samples. The quantitative results are included in Fig. 3a&3d. Note that our method covers all the 100 modes on the grid dataset while the best competitors LSGAN and VEEGAN cover 88 modes and 79 modes, respectively. Moreover, the number of modes covered by LBT increases consistently during the training. On the contrary, Unrolled GAN and VEEGAN can sometimes drop the covered modes, attributed to their unstable training.

Fig. 3b&3e show the results of averaged intra-mode KL divergence. We can see that LBT and LBT-GAN consistently outperform other competitors, which demonstrates that LBT framework can capture better intra-mode structure. According to Fig. 2c&2e, although LSGAN and VEEGAN can achieve good mode coverage, they tend to concentrate most of the density near the mode means and fail to capture the local structure within each mode. In LBT-GAN, the discriminator has a similar effect, while the estimator prevents the generator to over-concentrate the density. Therefore, the Intra-mode KL divergence of LBT-GAN may oscillate during training as in Fig. 3b.

Finally, we show the percentages of high quality samples for each method in Fig. 3c and Fig. 3f. We find that LBT-GAN achieves better results than LBT and outperforms other competitors. As LBT-GAN can benefit from the discriminator to generate high quality samples while maintaining the global and local mode coverage, we use LBT-GAN in the following experiments.

## 5.2 STACKED MNIST

Stacked MNIST (Metz et al., 2016) is a variant of the MNIST (LeCun et al., 1998) dataset created by stacking three randomly selected digits along the color channel to increase the number of discrete

---

[3]The exact expected number of high quality samples for each mode should be a little bit less than 62500 in this case, according to the three-sigma rule.

(a) CIFAR10: DCGAN(left) and LBT-GAN(right).          (b) CelebA: DCGAN(left) and LBT-GAN(right).

Figure 5: Generated samples on CIFAR10 (a) and CelebA (b) of DCGANs and LBT-GANs.

modes. There are 1,000 modes corresponding to 10 possible digits in each channel. Following (Metz et al., 2016; Srivastava et al., 2017), we randomly stack 128,000 samples serving as the training data. A classifier trained on the original MNIST data helps us identify digits in each channel. Following (Srivastava et al., 2017; Metz et al., 2016), we use 26,000 samples to calculate the number of modes to which at least one sample belongs. Besides, we also report the KL divergence between the generated distribution and the uniform distribution over the modes. Since reasonably finetuned GAN can generate 1000 modes, we select much smaller convolutional networks as both the generator and discriminator making our setting comparable to Metz et al. (2016). For LBT-GAN's VAE estimator, the number of hidden units of the two-hidden-layer MLP decoder and encoder are both 1000-400.

Table 1 presents the quantitative results. As we can see, LBT-GAN surpasses other competitors in terms of the number of captured modes, which demonstrates the effectiveness of the LBT framework. Specifically, LBT-GAN can successfully capture almost all modes under the LBT framework, and the results of KL divergence indicate that the distribution of LBT-GAN over modes is much more balanced compare to other competitors. Our method achieves comparable results with Pac-GAN (Lin et al., 2017). However, PacGAN is highly sensitive to the network architectures and it only generates 444 modes in our implementation, whereas LBT-GAN can successfully generalize to PacGAN's architecture and capture all 1000 modes. Our hypothesis is that the auxiliary estimator helps LBT generalize accross different architectures.

Fig. 4 shows the generated samples of GANs and LBT-GANs with different size of discriminators. The visual quality of the samples generated by LBT-GANs is better than GANs. Further, we find the sample quality of DCGANs is sensitive to the size of the discriminators, while LBT-GANs can generate high-quality samples under different network architectures.

### 5.3 CELEBA & CIFAR10

We also evaluate LBT on natural images, including CIFAR10 (Krizhevsky & Hinton, 2009) and CelebA (Liu et al., 2015) datasets. The generated samples of DCGANs and LBT-GANs are illustrated in Fig. 5. As the capacity of the vanilla VAE is not sufficient in such cases, LBT shows comparable results as the original GAN. We expect the performance of LBT can be boosted with much powerful estimators like Pixel-CNNs (Oord et al., 2016).

## 6 CONCLUSIONS & DISCUSSIONS

We present a novel framework LBT to train an implicit generative model via teaching an auxiliary likelihood estimator, which is formulated as a bilevel optimization problem. Unrolling techniques are adopted for practical optimization. Finally, LBT is justified both theoretically and empirically.

The main bottleneck of LBT is how to efficiently solve the bilevel optimization problem. For one thing, each update of LBT could be slower than that of the existing methods because the computational cost of the unrolling technique grows linearly with respect to the unrolling steps. For another, LBT may need larger number of updates to converge than GAN because training a density estimator is more complicated than training a classifier. Overall, if the bilevel optimization problem can be solved efficiently in the future work, LBT can be scaled up to larger datasets.

LBT bridges the gap between the training of implicit models and explicit models. On one hand, the auxiliary explicit models can help implicit models overcome the mode collapse problems. On the other hand, the implicit generators can be viewed as approximate samplers of the density estimators like Pixel-CNNs (Oord et al., 2016), from which getting samples is time-consuming. We discuss the former direction in this paper and leave the later direction as future work.

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

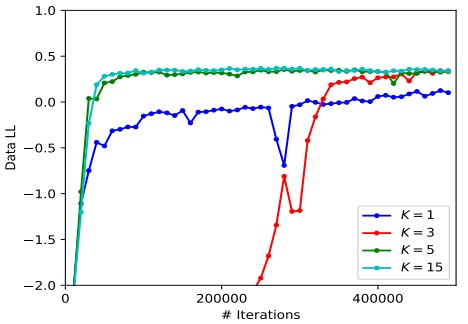 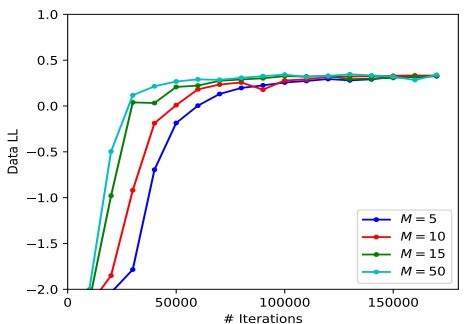

(a) Learning curves of LBT on the ring data with different unrolling steps $K$. The inner update iterations $M$ is fixed to 15.

(b) Learning curves of LBT on the ring data with different inner update iterations $M$. The unrolling steps $K$ is fixed to 5.

Figure 6: Sensitivity analysis of the unrolling steps $K$ and the inner update iterations $M$.

## A  SENSITIVITY ANALYSIS OF $K$ AND $M$

Theoretically, a larger unrolling steps $K$ can provide less biased gradients of $G$ and a larger inner update iterations $M$ can better approximate the condition in Eqn. (6) as analyzed in Sec. 3.1. However, large $K$ and $M$ on the other hand increase the computational costs. To successfully balance this trade-off, we provide sensitivity analysis of $K$ and $M$ in LBT. We adopt the values of the objective function Eqn. (1), i.e., the log-likelihood of real samples evaluated by the learned estimator, as the quantitative measurement.

We first investigate the influence of the number of unrolling steps $K$ on the training procedure. We use the experimental settings of the ring problem except that we vary the value of $K$. In Fig. 6a, we show the learning curves with $K = \{1, 3, 5, 15\}$. We observe that $K = 1$ leads to a suboptimal solution and larger $K$ leads to better solution and convergence speed. We do not observe significant improvement with $K$ larger than 5.

We show the influence of the number of inner update iterations $M$ during training in Fig. 6b. We use the experimental settings of the ring problem except that we use different $M = \{5, 10, 15, 50\}$. Our observation is that larger $M$ leads to faster convergence, which is consistent with the analysis in Sec. 3.1.

## B  AN ESTIMATOR WITH INSUFFICIENT CAPABILITY STILL HELPS THE GENERATOR

Below, we give a further analysis in the case where the estimator has limited capability. In this case, LBT itself cannot guarantee that the generator can model the data distribution since the nonparametric assumption in Sec. 4 is not met, but LBT still provides complementary information to GAN and can improve GAN. Below, we empirically verify this argument with two examples of LBT-GAN:

1. a toy example where an estimator with insufficient capability can help $G$ escape a bad local optimum of GAN;

2. the Stack-MNIST experiment where a much smaller VAE can help $G$ cover 1000 modes and generate realistic samples.

For the toy example, we consider the same settings as in Sec. 1. Suppose that the data distribution is a mixture of Gaussians (MoG), i.e., $p(x) = 0.5\mathcal{N}(-3, 1) + 0.5\mathcal{N}(3, 1)$ and the generator $G$ is also a MoG, i.e., $p_G(x) = 0.5\mathcal{N}(a, 1) + 0.5\mathcal{N}(b, 1)$ which has enough capacity to capture $p(x)$. In this case, it is possible for GAN to fall into bad local optima that collapse to certain mode, e.g., $a = -3.25, b = -2.69$, because of the property of JS-divergence. In LBT, we suppose that the estimator $E$ is with insufficient capability, i.e., a single Gaussian $p_E(x) = \mathcal{N}(c, 1)$ with a learnable mean $c$, which can only capture the mean of a distribution. In this case, $G$ needs to teach $E$ to

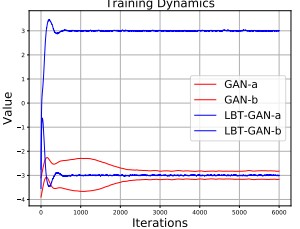 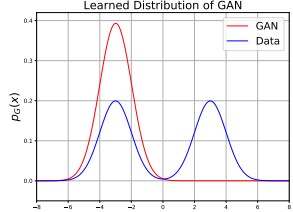 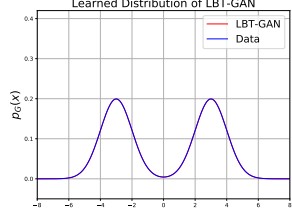

(a) The training process of both GAN and LBT-GAN with a toy example.

(b) The learned distribution of GAN (left) and LBT-GAN (right).

Figure 7: The training process and learned distribution of GAN and LBT-GAN for a simple toy example.

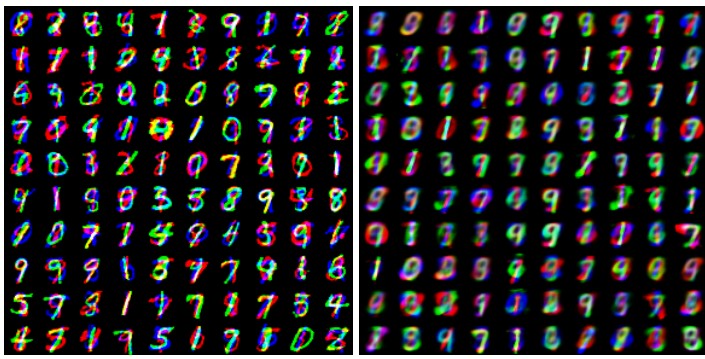

Figure 8: The generated samples of LBT-GAN with a smaller VAE (left) and the samples from the smaller VAE (right).

capture the mean of $p(x)$, i.e., $c = 0$. Though $E$ does not capture $p(x)$, it still regularizes $G$ to learn a $p_G(x)$ with zero mean. Therefore, if $p_G$ is around the bad local optimum of GAN whose mean is about $-2.97$ (where the gradients of the GAN part of LBT-GAN will be nearly zero), the gradients of the LBT part will encourage the generator to overspread the data distribution. Namely, even given an estimator of limited capability, LBT-GAN has no such bad local optima thanks to the LBT part. A clear demonstration is shown in Fig. 7, where we identically initialize the means of $G$ in both LBT-GAN and GAN around $-3$. GAN converges to the local optimum of JS-divergence, whereas LBT-GAN can successfully regularize the generator to a distribution with zero mean and converges to the global optimal quickly.

For the second example, we re-implement our LBT-GAN on the Stacked-MNIST dataset with a much smaller VAE where both the encoder and the decoder are two-hidden-layer MLPs with only 20 units in each hidden layer. The samples from the smaller VAE are of poor quality (See the right panel of Fig. 8), which means that it can hardly capture the distribution of Stacked-MNIST. Nevertheless, using such a simple VAE, LBT-GAN can still cover all the 1000 modes and generate visually realistic samples which is illustrated in the left panel of Fig. 8.

