# OpenReview forum: "Learning Implicit Generative Models by Teaching Explicit Ones"
_ICLR.cc/2019/Conference_

### Official Review · AnonReviewer1 · 2018-11-02
**Good cocktail of ideas, limited analysis**

**Rating:** 6
**Confidence:** 4

**Review:**

This work introduces a framework for learning implicit models that is robust to mode collapse. It consists in learning an explicit model of the implicit model through maximum likelihood while the later is used to teach the explicit model to better match the data distribution. The resulting bi-level optimization is carried out with truncated unrolled stochastic gradient descent.

# Quality

The method combines an interesting set of ideas. It is validated on some reasonable experiments.

However after reading the paper, I remain with too many unanswered questions:
- Why should the method avoid mode collapse? Experiments clearly show that it indeed is resilient to mode collapse, but I have would have been curious in seeing some more discussion regarding this point. What is the exact mechanism that solves the issue?
- What is the effect of K? Is mode collapse solved only because of the unrolled gradients?
- What is the effect of M? How does the method behave for M=1, as usually done in GANs?
- What if the explicit model has not enough capacity?
- The original Unrolled GAN paper presents better results for the ring problem. Why are results worse in the experiments?

More fundamentally what is the main benefit of this approach with respect to models that can be trained straight with maximum likelihood? (e.g., flow-based neural generative models; and as required for the explicit model) Is it only to produce generative models that are fast (because they are implicit)? Why not training only the explicit model directly on the data?

# Clarity

The paper is in general well-written, although some elements could be removed to actually help with the presentation.
- The development around influence functions could be removed, as the method ends up instead making use of truncated unrolled gradients.
- The theoretical analysis is straightforward and could be compressed in a single paragraph to motivate the method.

# Originality

The method makes use of several ideas that have been floating around and proposed in different papers. As far as I know, the combination proposed in this work is original.

# Significance

Results show clear resistance to mode collapse, which is an improvement for implicit models. However, other types of generative models generally do not suffer from this issue. Significance is therefore limited.

---

> ### Author Response · Authors · 2018-11-21
> **Response to Reviewer 1**
>
> Thanks for your valuable comments and questions. Below, we give a further explanation on the mechanism on why LBT can avoid mode collapse, and provide sensitivity analysis on the hyperparameters of $M$ and $K$. The questions are answered following the line order.
>
> Q1: Why should the method avoid mode collapse? Experiments clearly show that it indeed is resilient to mode collapse, but I have would have been curious in seeing some more discussion regarding this point. What is the exact mechanism that solves the issue?
> A1: Please refer to our post with title “Response to a common concern: What’s the mechanism of LBT that solves mode collapse?”.
>
> Q2: What is the effect of K? Is mode collapse solved only because of the unrolled gradients?
> A2: Theoretically, when $K=\infty$, the gradients of generator parameters obtained by the unrolling technique are unbiased regardless of the initial point of the estimator. And if the estimator achieves optima, only one step of unrolling, i.e., $K=1$, is enough, according to the analysis in Sec. 3.1. Generally speaking, a larger $K$ provides less biased gradients for $G$ w.r.t. the initial point of the estimator, which on the other hand needs more computational resources. In our experiments, we find $K=5$ is enough to provide valid gradients to the generator. We verify this argument with sensitivity analysis for $K$ in Appendix A of our revised paper and demonstrate that a larger $K$ leads LBT to a better optimum.
> For the second question, as mentioned in our post with title “Response to a common concern: What’s the mechanism of LBT that solves mode collapse?”, it is KL divergence rather than the unrolled gradients that solves mode collapse.
>
> Q3: What is the effect of M? How does the method behave for M=1, as usually done in GANs?
> A3: We’d like to clarify that the roles of the hyperparameter $M$ in LBT and GAN are different. In LBT, a larger $M$ can better approximate the condition in Eqn. (6) as analyzed in Sec. 3.1 (See Appendix A of the sensitivity analysis.). In GAN, $M$ balances the training of $G$ and $D$, which needs to be carefully tuned [1].
>
> Q4: What if the explicit model has not enough capacity?
> A4: Please refer to our post with title “Response to a common concern: Can an estimator with insufficient capability still help the generator to capture the data distribution?”.
>
> Q5: The original Unrolled GAN paper presents better results for the ring problem. Why are results worse in the experiments?
> A5: We produce the results of the ring problem following Unrolled GAN’s settings except the true data distribution. We use the ring data where the standard deviation (std) of each component is 0.1 and the radius of the ring is 1; In the original Unrolled GAN’s setting, the std of each component is 0.02 and the radius of the ring is 2. Note that the ratio of std to radius in our setting is 10 times larger than in the original Unrolled GAN’s setting, making the ring problem more difficult. We choose this setting in order to characterize different performance of "Intra-mode KL divergence" clearly. In fact, we also tried the original Unrolled GAN’s setting of the ring problem. We find similar performance of all methods and did not find mode collapse except the vanilla GAN. We added the discussion about the different settings for the ring problem in Sec. 5.1.
>
> Q6: What is the main benefit of this approach with respect to models that can be trained straight with maximum likelihood?
> A6: Maximum likelihood estimation is applicable to the models with an explicit likelihood. It has been widely shown that implicit models have many attractive properties compared to explicit models, such as the model flexibility and the ability to generate sharp and realistic images [2] efficiently. Concretely, implicit models can generate realistic images compared to VAEs, and can generate samples efficiently compared to PixelCNN. Compared to flow-based methods, implicit models can define a more flexible distribution whereas the generator network in flow-based methods has to be invertible which largely limits the capacity.
> Our method proposes a new framework to train implicit models, and bridges the gap between the success of explicit models and the challenge of learning implicit models.
>
> [1] Berthelot, David, Thomas Schumm, and Luke Metz. "BEGAN: boundary equilibrium generative adversarial networks." arXiv preprint arXiv:1703.10717 (2017).
> [2] Mohamed, Shakir, and Balaji Lakshminarayanan. "Learning in implicit generative models." arXiv preprint arXiv:1610.03483(2016).

---

### Official Review · AnonReviewer2 · 2018-11-03
**Avoiding Mode Collapse?**

**Rating:** 5
**Confidence:** 4

**Review:**

This paper presents an learning by teaching (LBT) framework to train implicit generative models. Instead of using discriminator as in GANs, LBT adopts an explicit likelihood estimator as the student, which is formulated as a bilevel optimization problem:
1) maximize the log-likelihood of the generated samples;
2) maximize the log-likelihood evaluated on the training data.
The authors argue that LBT avoids the mode collapse problem intrinsically as the missing modes will be penalized by the second objective. I have some concerns on this.  Why teaching an explicit likelihood can help learn an implicit one?

Suppose the explicit likelihood estimator is a single Gaussian, but the real distribution has multiple modes, fitting such the generated data and the training data on this likelihood will not help to avoid missing modes.

From the empirical results, it is clear that LBT-GAN is better than LBT. From the objective in (8), it seems the true reason is  the P_E and D together representing a mixture model, which may fit the training data better.

In Figure 2.(b), the Intra-mode KL divergence of LBT-GAN seems to be unstable during the training, is this caused by the joint training of discriminator with the estimator. Can you discuss this?

In Table 1, the authors just copied the results of VEEGAN. Indeed, in our implementation, DCGAN and VEEGAN can be much better than the reported one. The authors have not tried the effort to tune the results of baselines.

Recently, the Least square GAN has been purposed to address the mode collapse as well. I suggested the authors should empirically compare with it as well.

Generally, the paper is well-written. The idea is interesting, however, the motivation, analysis and empirical results are not convincing enough to fully support their claim.

---

> ### Author Response · Authors · 2018-11-21
> **Response to Reviewer 2 (Part 1)**
>
> Thank you for the valuable comments. Below, we address the concerns following the line order. In the revised paper, we have clarified the motivation, analyzed the mechanism for solving mode collapse in Sec.1 and included more empirical results in Sec.5 and Appendix A and B.
>
> Q1: Why teaching an explicit likelihood can help learn an implicit one?
> A1: Please refer to our post with title “Response to a common concern: What’s the mechanism of LBT that solves mode collapse?”.
>
> Q2: “Suppose the explicit likelihood estimator is a single Gaussian, but the real distribution has multiple modes, fitting such the generated data and the training data on this likelihood will not help to avoid missing modes.”
> A2: In this case, LBT itself cannot guarantee that the generator can model the data distribution, but LBT-GAN can surpass GAN thanks to the LBT part. In our post with title “Response to a common concern: Can an estimator with insufficient capability still help the generator to capture the data distribution?”, we provide detailed analysis using a toy example and more results on the Stacked-MNIST dataset.
>
> Q3: “From the empirical results, it is clear that LBT-GAN is better than LBT. From the objective in (8), it seems the true reason is the P_E and D together representing a mixture model, which may fit the training data better.”
> A3: We clarify that $p_E$ and $D$ do not represent a mixture model, since $p_E$ is a density function while $D$ is not. In fact, it is the generator $G$ rather than $E$ and $D$ that fits the training data in LBT-GAN. (Note that the estimator $E$ is always trying to fit the generated samples.) The way how $E$ and $D$ work together is that $E$ and $D$ together contribute to the gradient of $\theta$ (generator parameters).
> We’d like to emphasize that the roles of $E$ and $D$ in LBT-GAN are quite different, as mentioned in the post with the title “Response to a common concern: Can an estimator with insufficient capability still help the generator to capture the data distribution?”. In LBT-GAN, $E$ supervises $G$ to overspread the support of data distribution, and $D$ penalize the fake patterns to generate realistic samples. Therefore, LBT and GAN are complementary to each other, implying that LBT-GAN outperforms not only GAN but also LBT.
>
> Q4: In Figure 2.(b), the Intra-mode KL divergence of LBT-GAN seems to be unstable during the training, is this caused by the joint training of discriminator with the estimator. Can you discuss this?
> A4: In the ring experiments, in order to distinguish the different performance on “Intra-mode KL divergence”, we choose the ring data distribution with larger standard deviation for each component (Please also see Q5 to Reviewer1). Within such “larger” components, we clearly observe “intra mode collapse” for GAN based methods — they tend to concentrate most of the density near the mode means and fails to capture the local structure within each mode — such as in VEEGAN (Fig. 1e) and LSGAN (Fig. 1c). In LBT-GAN, the discriminator has a similar effect, while the estimator has the effect to prevent the generator to over-concentrate the density. Therefore, the Intra-mode KL divergence of LBT-GAN may oscillate. For the grid problem, since the standard deviation for each component is much smaller, no clear “intra mode collapse” is observed and thus the Intra-mode KL divergence is relatively stable. We have added the discussion in Sec.5.1 in the revised paper.

---

> ### Author Response · Authors · 2018-11-21
> **Response to Reviewer 2 (Part 2)**
>
> Q5: In Table 1, the authors just copied the results of VEEGAN. Indeed, in our implementation, DCGAN and VEEGAN can be much better than the reported one. The authors have not tried the effort to tune the results of baselines.
> A5: Indeed, we have tried our best to tune the results of VEEGAN. As stated in Unrolled GAN [1], using a large model with fine-tuned hyper-parameters, a simple DCGAN can cover 1000 modes. So we follow the most direct competitor [1] and use G and D with a limited number of parameters to demonstrate the robustness of the proposed methods (See Sec. 5.2). Since the authors of VEEGAN did not release the source code and hyperparameters on the Stacked MNIST dataset, we tried our best to implement VEEGAN and compare it with LBT-GAN, Unrolled GAN and PacGAN in the same setting. However, though we carefully fine-tuned the hyperparameters of VEEGAN according to the released code of VEEGAN for the toy data, the generator always collapsed to a single pattern. This may be because VEEGAN is more sensitive to the hyperparameters than LBT-GAN. We further provide an additional sensitivity analysis about the hyperparameters in LBT in Appendix A of our revised paper, which shows that LBT is more robust to the hyperparameters. Finally, we will release our source code on all these methods after the blind review.
>
> Q6: Recently, the Least square GAN has been proposed to address the mode collapse as well. I suggested the authors should empirically compare with it as well.
> A6: Thanks for your suggestion. We have included the results of LSGAN in the revised paper (See Fig. 1 & Fig. 2). We observe that LSGAN can achieve better mode coverage (88 modes in the grid problem) than other baselines. However, like VEEGAN, LSGAN tend to collapse most of the density to the mode centers and fail to capture the local structure within each mode.
>
> [1] Metz, Luke, et al. "Unrolled generative adversarial networks." arXiv preprint arXiv:1611.02163 (2016).

---

### Official Review · AnonReviewer3 · 2018-11-03
**Well-written,  interesting experiments, doubts about scalability**

**Rating:** 7
**Confidence:** 3

**Review:**

The authors present a novel architecture of  an implicit unsupervised learning architectures using
a teacher student approach.  In particular the main advantage to me seems to be the mode-collapse property,  an important drawback in standard
GAN approaches.

The paper is written very well and is easy to follow. The methodology is presented in a clear way and the experiments make sense given the research question.  I particular like that the authors define clear metrics to evaluate success, which is often the weak point in unsupervised learning problems.

I believe the work is interesting, but the results still preliminary and  possibly limited by  scalability.  As the authors put it

"The main bottleneck of LBT is how to efficiently solve the bi-level optimization problem. On one
hand, each update of LBT could be slower than that of the existing methods because the computational
cost of the unrolling technique grows linearly with respect to the unrolling steps."

On the other hand, I appreciate the honesty in discussing possible scalability constraints.

I was a bit surprised that the method the authors propose seems to work well in the  "Intra-mode KL divergence".  My expectation was  that the main advantage of your method is capturing the global, holistic shape of the distribution
of the data, whereas classical methods would, because of mode collapse, only capture specific  sub-spaces.  Therefore, i would expect these classical methods to perform better in intra-mode KL divergence,  which is a metric to measure local
, not global, approximation quality.

Typos:
-  In practise (Introduction) -> in practice
- 3.1 accent -> ascend
- Conclusion: on one hand / other hand is used for two opposite ways of thinking

---

> ### Author Response · Authors · 2018-11-21
> **Response to Reviewer 3**
>
> Thank you for your comments. We have addressed these typos in our revised paper. Besides, we provide further analysis about the advantages of LBT and LBT-GAN over GAN in terms of resisting mode collapse. Please refer to our posts about two common concerns and the revised paper.

---

### Author Response · Authors · 2018-11-21
**Response to a common concern: What’s the mechanism of LBT that solves mode collapse?**

We clarify that LBT is more resistant to mode collapse than GAN because of the intrinsic difference between the statistical divergences used. The KL-divergence in LBT encourages the generator $G$ to cover more modes compared to the JS-divergence in GAN. We verify this argument with a toy example below and provide an intuitive explanation in Sec. 1 of the revised paper.

Suppose that the data distribution is a mixture of Gaussian (MoG), i.e., $p(x) = 0.5N(-3, 1) + 0.5N(+3, 1)$, and the generator distribution is a MoG with two learnable means $a$ and $b$, i.e., $p_G(x) = 0.5N(a, 1) + 0.5N(b, 1)$, indicating the generator has enough capacity to capture the data distribution. Under such condition, $a=-3.25, b=-2.69$ (numerical results) is a local optimum of the JS-divergence $JSD(p||p_G)$, which misses the mode with mean $+3$. In contrast, $KL(p||p_G)$ achieves its optima iff $p=p_G$ in this case, which indicates that the KL-divergence is more resistant to mode collapse. (The contours of the JS-divergence and the KL-divergence of the above example can be found in Sec. 1 (Introduction) of our revised paper.) With a poor initialization and gradient-based optimization methods, the JS-divergence may lead to the local optimum mentioned above. We empirically verify this argument in Appendix B (Fig. 7(a)), where the two red curves show the training dynamics of $a$ and $b$.

Therefore, given sufficient model capability, it is possible that vanilla GAN falls into certain local optimum with mode collapse while LBT does not. In our experiments, we also empirically show that LBT performs better than various existing methods to improve GAN.

---

### Author Response · Authors · 2018-11-21
**Response to a common concern: Can an estimator with insufficient capability still help the generator to capture the data distribution?**

In our post “Response to a common concern: What’s the mechanism of LBT that solves mode collapse?”, we explain how an estimator with sufficient capability helps our generator to successfully capture the data distribution. Below, we give a further analysis in the case where the estimator has limited capability. In this case, LBT itself cannot guarantee that the generator can model the data distribution since the nonparametric assumption in Sec. 4 is not met, but LBT still provides complementary information to GAN and can improve GAN. Below, we empirically verify this argument with two examples of LBT-GAN:
(1) a toy example where an estimator with insufficient capability can help $G$ escape a bad local optimum of GAN;
(2) the Stack-MNIST experiment where a much smaller VAE can help $G$ cover 1000 modes and generate realistic samples.

For the toy example, we consider the same settings as in the post “Response to a common concern: What’s the mechanism of LBT that solves mode collapse?”. Specifically, suppose that the data distribution is a mixture of Gaussians (MoG), i.e., $p(x) = 0.5N(-3, 1) + 0.5N(+3, 1)$ and the generator $G$ is a MoG with two learnable means $a$ and $b$, i.e., $p_G(x) = 0.5N(a, 1) + 0.5N(b, 1)$. In this case, as shown in the post “Response to a common concern: What’s the mechanism of LBT that solves mode collapse?”, GAN can possibly fall into bad local optima, e.g., $a=-3.25, b=-2.69$, which collapses to certain mode.
In LBT, we consider a simple estimator $E$ with insufficient capability, i.e., a single Gaussian $p_E(x)=N(c, 1)$ with a learnable mean $c$, which can only capture the mean of a distribution. In this case, $G$ needs to teach $E$ to capture the mean of $p(x)$, i.e., $c=0$. Though $E$ does not capture the true data distribution, it still regularizes $G$ to learn a $p_G(x)$ with zero mean. Therefore, if $p_G$ is around the bad local optimum of GAN whose mean is $-2.97$ as we mentioned above (the gradients of the GAN part of LBT-GAN will be nearly zero), the gradient of the LBT part will encourage the generator to overspread the data distribution. Namely, even given an estimator of limited capability, LBT-GAN has no such bad local optima thanks to the LBT part. We show more details, including the learning dynamics and the learned density functions from different models, in Appendix B.

For the second example, we re-implement our LBT-GAN on the Stacked-MNIST dataset with a much smaller VAE where both the encoder and the decoder are two-hidden-layer MLPs with only 20 units in each hidden layer. The samples from the smaller VAE are of poor quality, which means that it can hardly capture the distribution of Stacked-MNIST. In this setting, LBT-GAN can still recover 1000 modes and generate realistic samples. Related results are updated in Appendix B.

In conclusion, LBT still provides complementary information to GAN and can improve GAN given an estimator with limited capability.

---

### Meta-Review · Area_Chair1 · 2018-12-08
**Intersting paper that would profit from a better understanding of the underlying mechanism**

**Confidence:** 3
**Recommendation:** Reject

**Metareview:**

The paper proposes a learning by teaching (LBT) framework to train an implicit generative model via an explicit one. It is shown experimentally, that the framework can help to avoid mode collapse. The reviewers commonly raised the question why this is the case, which was answered in the rebuttal by pointing to the differences between the KL- and the JS-divergence and by showing a toy problem for which the JS-divergence has local minima while the KL-divergence has not. However, it still remains unclear why this should be generally and for explicit models with insufficient capacity the case, and if the model will be scalable to larger settings, therefore the paper can not be accepted in the current form.